# Comparison of Pyrene Biodegradation Using Two Types of Marine Bacterial Isolates

Ismail Marzuki [1,*], Khairun Nisaa [2], Ruzkiah Asaf [2], Admi Athirah [2], Mudian Paena [2], Endang Susianingsih [2], Nurhidayah Nurhidayah [2], Ince Ayu Khairana Kadriah [2], Kamaruddin Kamaruddin [2], Sahabuddin Sahabuddin [2], Nurbaya Nurbaya [2], Early Septiningsih [2], Herlinah Herlinah [2], Erfan Andi Hendrajat [2], Suwardi Suwardi [2] and Andi Ramlan [3]

1   Department of Chemical Engineering, Fajar University, Makassar 90231, Indonesia
2   National Research and Innovation Agency (BRIN), Central Jakarta 10340, Indonesia
3   Marine and Coastal Resources Management Agency of Makassar, Makassar 90512, Indonesia
*   Correspondence: ismailmz@unifa.ac.id

**Abstract:** Polycyclic aromatic hydrocarbons (PAHs) contaminants have toxic, carcinogenic, and mutagenic properties. Screening bacteria from different sources capable of carrying out the biodegradation of (PAHs) is essential for mapping and mobilization purposes and applying them to polluted hydrocarbon environments. The study aims to compare the capacity of PAH biodegradation by two types of bacteria isolated from different sources. The method applied is the interaction between bacterial suspension and pyrene-contaminated waste for 30 days. Biodegradation products in organic compounds were analyzed using gas chromatography/mass spectroscopy (GC/MS) and Fourier transform infrared spectroscopy (FTIR). The analysis results found several indications of the performance of bacterial biodegradation: The capacity of pyrene degradation by *Bacillus licheniformis* strain ATCC 9789 (Bl) bacteria against pyrene was relatively more dominant than *Sphingobacterium* sp. strain 21 (Sb) bacteria. The percentage of total bacterial biodegradation for product type Sb was (39.00%), and that of the product of bacterial degradation type Bl (38.29%). The biodegradation products of the test bacteria (Bl and Sb) were relatively similar to pyrene in the form of alcohol and carboxylic acid organic compounds. There was no significant difference in the pyrene biodegradation between Bl and Sb bacteria.

**Keywords:** biodegradation; pyrene; pollutants; bacteria; marine sponges; polluted seawater

## 1. Introduction

Polycyclic aromatic hydrocarbons (PAHs) are non-polar molecules. The structure of a PAH is composed of carbon and hydrogen atoms, and has no charge, a typical characteristic because the ring structure is capable of delocalizing electrons in the aromatic ring [1–4]. PAH components in nature are naturally available, and generally found in coal, petroleum, and organic materials that have undergone thermal decomposition [5–7]. Population, dynamics, and efforts to meet the needs of human life have resulted in increasing exploration and exploitation of fossil deposits, coal, and decomposed organic biomass, resulting in the potential for disposal of PAHs components in nature to increase yearly [4,8–10]. Aromatic hydrocarbon chemicals consist of several types [11–13]. The simplest have two aromatic rings such as naphthalene or three rings (anthracene and phenanthrene), while pyrene has four aromatic rings forming a stable structure, and other PAHs have additional rings [1,14–16]. PAHs generally have toxic and even carcinogenic and mutagenic properties [17–19]. The toxic level of PAHs tends to increase as the structure of the aromatic ring member increases. Pyrene is one of the PAHs with a relatively high level of toxicity [2,20,21].

The effects caused by exposure to pyrenes and similar PAHs on marine ecosystems need to be monitored because the sea is a giant container that provides space

for almost all waste materials on earth, including several types of dangerous and toxic contaminants [9,22–24]. Such exposure can lead to a chain effect, which can cause problems and impact human health [3,25–27]. However, marine life has materials that can reduce, degrade, and deactivate the toxic properties of PAHs components, especially microorganisms, such as bacteria and fungi [5,28–30].

Several types of bacteria are part of the marine biota life that can degrade hydrocarbon pollutants [19,31,32]. The mutualistic symbiosis of bacteria with sponges is a common finding. Sponges are marine biotas that have life dynamics as filter feeders, often objects and materials for research studies in biomonitoring and bioindicators of the pollution components of heavy metals and PAHs [21,33–35]. This situation indicates that sponges can adapt or survive in environments exposed to PAHs and heavy metals [7,36,37]. Sponges are in symbiosis with microorganisms, especially bacteria, so conducting an in-depth study of whether there is a relationship between sponges, bacterial symbiosis, and PAHs contaminants would be of interest [17,38–40].

Many research reports show that several types of bacteria can degrade hydrocarbon components, where bacteria can absorb carbon and convert it into energy [41–43]. Bacillus and pseudomonas bacteria are known to be able to carry out the function of biodegradation of hydrocarbon components [4,44,45]. Gram-positive bacteria, found in the form of bacilli, generally have aerobic properties, and some can become anaerobic when oxygen is unavailable [3,46,47]. Bacillus group bacteria that live in contaminated environments can produce endospores as a form of camouflage and can survive for long periods.

Bacteria of the bacillus group often exhibit symbiosis with marine sponges [48]. The Sphingobacterium group of bacteria is a genus that belongs to the *Sphingobacteriaceae* family, containing high concentrations of *sphingophospholipids* [11,49]. *Sphingobacterium* is a group of bacteria isolated from several habitats, one of which can be obtained from seawater [50–52]. Both groups of bacteria can biodegrade aromatic hydrocarbon components. The main objective of this research is the availability of quantitative data related to the biodegradation strength of a type of bacteria obtained from different sources [53–55].

Bioremediation of PAHs using microorganisms has been widely developed. Several types of bacteria capable of carrying out biodegradation of PAHs isolated from marine water are suspected to be contaminated with hydrocarbons, including *Bacillus* [56,57], *Gammaproteobacteria*, and Pseudomonadales [4,58]. Isolates from hydrocarbon contaminated soil include *Micrococcus luteus* [59], *Lasiodiplodia theobromae* [59–61], and Pseudomonas aeruginosa [62]. Several bacteria were isolated from marine biota, especially marine sponge microsymbionts, for example, *Bacillus* sp. strain AB353f, *B. pumilus* strain GLB197, *B. cohnii* strain DSM 6307, and *Acinetobacter Calcoaceticus* strain PHCDB14 [9,63–65]. Microorganisms associated with mangroves have also been identified to carry out the biodegradation function of PAHs [40,66,67]. Their research includes the biodegradation of hydrocarbon components using *Ganoderma lucidum*, Penicillium sp., and *filamentous* isolated from fungi [3,7,32,68]. Types of PAHs that have been successfully degraded by a number of microorganisms, including bacteria and fungi, include naphthalene [8,21,69], anthracene [4,10,70], phenanthrene [7,71], pyrene [2,4,7,40,72], and benzo(a)pyrene [36,37,73].

Comparative analysis of the strength of bacterial biodegradation of PAH components is important in mapping the type, source, and effectiveness of the biodegradation of these bacteria against PAHs [74–76]. The data from this research can also be used to develop and apply environmental bioremediation against other types of pollutants, such as heavy metals, microplastics, and pesticide residues. Future bioremediation using bacteria is likely to occur in treating medical waste, radioactive, and other hazardous chemicals [6,77–79]. The use of bacteria that have remediation capabilities can also potentially be applied to liquid waste, solid waste environments, and even air. The development of knowledge on screening the type and source of bacteria is attractive, especially for microorganisms such as fungi [44,57,80].

Research on the biodegradation of pollutants in the environment using microorganisms such as bacteria and fungi is still an open topic. It is an interesting research area

with potential future benefits in environmental management [81–83]. This research is part of a series and developments of several previous studies on carcinogenic PAHs with the theme of screening for hydrocarbon degrading bacteria [12,84]. The novelty presented in this research is the source of bacteria isolated from marine sponges for the application of biodegradation of PAHs [85,86].

Polycyclic aromatic hydrocarbon (PAHs) pollution can harm marine ecosystems and human safety due to its bioaccumulation, biomagnification and biodegradability potential, toxicity, and toxicity and carcinogenicity in nature [87,88]. Thus, the importance of degrading these pollutants from aquatic environments by an ecological method such as biodegradation using bacteria isolated from suitable environments capable of decomposing them is undeniable [3,4,86]. Based on this, it is necessary to make efforts to reduce PAHs contaminants by applying one of the methods, namely a combination of microbiological studies with determination of the level of pyrene degradation using GC/MS and analysis of biodegradation product functional groups using FTIR, which allows for an excellent characterization of the pyrene biodegradation scenario and evaluation of the tested bacterial strains under a condition for the performance of appropriate tests and proper interpretation of the results [89–92].

## 2. Materials and Methods

### 2.1. Materials and Bacterial Strains

The material used is *Bacillus licheniformis* strain ATCC 9789 (Bl), *Sphingobacterium* sp. strain 21 (Sb), pyrene for GC as the main ingredient in the analytical standard (Supelco-Sigma Aldrich, Santa Clara, CA, USA), and other materials, such as N-hexane (brand) for GC, anhydrous $Na_2SO_4$, ethanol, peptone, glucose, nutrient agar, physiological NaCl 0.9%, yeast extract, aquabides, ethanol, and nitrogen gas. The gas chromatography/mass spectrometer (GC/MS) from Agilent Type 7890A (operating conditions for GC/MS max) is the main instrument for measuring biodegradation performance. The GC/MS operating temperature is 350 °C. The operating temperature of the instrument is increased slowly, i.e., the temperature rise is 10 °C every 5 min and the pressure is 18,406 psi. To work stably, a Helium gas carrier, a speed of 150 mL/min, capillary column (Agilent 19019S-436HP-5 ms), (Sigma-Aldrich, Santa Clara, CA, USA) and Fourier transform infra-red (FTIR) Shimadzu IR Prestige-21 and OD600 spectrophotometer visible [11,49,90–92].

Two types of bacteria were used, Bl and Sb. Bacterial Bl was isolated from the marine sponge *Auletta* sp. around Kodingareng Keke Island, a small island included in the Marine Tourism Area of Makassar City, Spermonde Archipelago Cluster (Figure 1A). *Bacterial* Sb was isolated from marine water suspected to be contaminated with hydrocarbon components, precisely around Soekarno Hatta port (Figure 1B) [4,66]. The sampling point distance of the two locations (Figure 1A,B) is approximately 21 km. The selection of these two types of bacteria (Bl and Sb) from different sources was based on data on phenotypic characterization using standard 16-parameter biochemical reagents and data on genotypic analysis using PCR that researchers had previously carried out. Bacterial isolates Bl and Sb are isolates of the research stock [4,6,89,90].

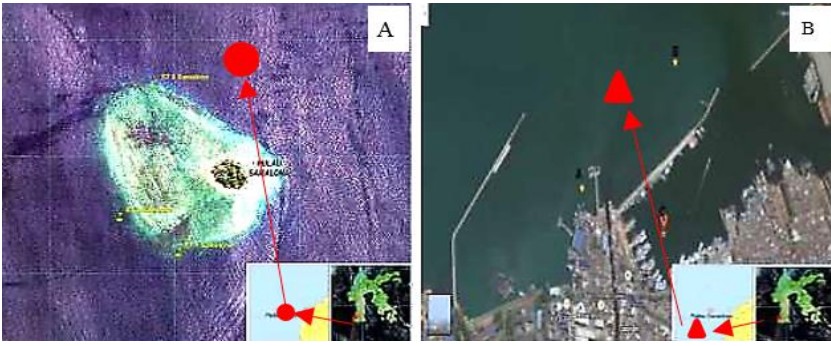

**Figure 1.** Sampling locations of sea sponges and seawater sources of bacterial isolates. The distance between the two sampling points is ±3.45 km; (**A**) map of Kodingareng Keke Island, part of the Spermonde Archipelago Cluster, Makassar City Administration. The sampling location of the marine sponge *Auletta* sp., the source of Bl. Sponge sampling point (red circle) at coordinates 5°06′38.12′76″ S; 119°17′7.76′44″ E; (**B**) sampling location of marine water source of bacteria type Sb (red triangle), obtained around Soekarno-Hatta Seaport, Makassar. The seawater sampling point is at coordinates 5°06′25.23′14″ S; 119°25′3,21.25″ E.

## 2.2. Sample Preparation

In the first stage, bacterial isolate cells were propagated using the culture method. The culture was carried out in a test tube, and then the bacterial cells were suspended using aquabides. Incubation for $1 \times 24$ h was carried out along with a Gram staining test to confirm the Gram groups of the two types of bacteria tested. A row of sterilized labeled degradation vials was prepared. Each vial was filled with 10 mL of bacterial suspension and then adapted to the new environment for $1 \times 24$ h in an incubator. In the second stage, 200 mL of pyrene 1000 mg/L was made [6,9,21]. In each degradation vial that already contained a bacterial suspension, 5 mL of pyrene solution was added so that the interaction between the bacterial suspension and the pyrene solution occurred.

## 2.3. Performance of Bacteria and Biodegradation Products

Each degradation vial was placed in a shaker incubator and agitated at 200 rpm. The contact between the bacterial suspension and pyrene (substrate) lasted for 30 days. Every 3 days, biodegradation parameters (optical density) were observed and measured. Measurement of the level of biodegradation was carried out after the interaction time of 10, 20, and 30 days using GC/MS [4,19,23,89].

The determination of the level of biodegradation was carried out using all samples in the vial that had reached an interaction period of 10 days, and their multiples were extracted using N-hexane to extract the pyrene component that was not degraded. The N-hexane extract was added with $Na_2SO_4$ to attract water components and other contaminants that could interfere with the measurement using GC/MS. The N-hexane extract was then used to obtain data on the performance of bacteria and components of biodegradation products via GC/MS [4,90,91]. The N-hexane extract was also used to obtain data on the types of components of the biodegradation product using FTIR, according to the functional groups shown on the chromatogram [14,68,91,92].

## 3. Results

### 3.1. Morphological Analysis

The culturing process of two isolates used as PAH degradators was carried out using different sources. *Bacillus licheniformis* strain ATCC 9789 (Bl) was isolated from the marine sponge *Auletta* sp., while *Sphingobacterium* sp. strain 21 (Sb) was isolated from marine water and was suspected of being exposed to PAHs. The two test bacteria were selected based on the biodegradation potential of their PAHs in previous studies [3,5,93]. The process of

culturing, morphology, and microscopy of the two types of isolates used to degrade PAHs, especially pyrene, is shown in Figure 2.

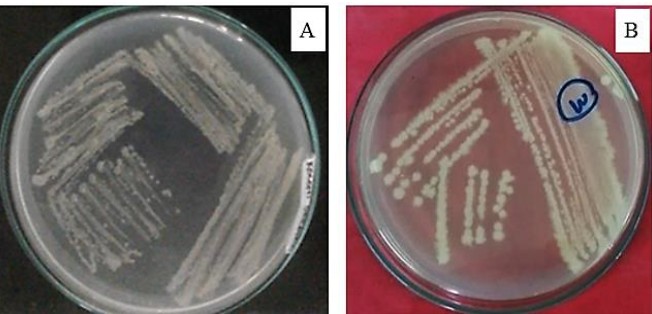

**Figure 2.** Comparison of morphology of two types of bacterial isolates cultured on NA selective media, after incubation 1 × 24 h; (**A**) growth of type Bl bacterial isolates; (**B**) growth of Sb type bacterial isolates.

According to the criteria for phenotypic and genotypic characteristics, isolates Bl and Sb were selected. Both isolates were cultured, and the culture results were converted into a suspension. The microscopic analysis results show that isolate Bl was isolated from marine sponge type *Auletta* sp., and the sponge was obtained around Kodingareng Keke Island (Figure 1A) [30,50,94].

The morphology of bacterial isolate Bl (Figure 2A) can be illustrated as follows: ridged rod shape, cream color, spread in clusters, endospores, and less clear, while bacterial isolate Sb (Figure 2B) has a ridged rod shape, brown color, different distribution, and lack an endospore [41,50]. Thus, it is suspected that there are differences in the degradation ability of PAH components, especially pyrene [4,19,39]. Comparison of growth rates between Bl and Sb bacteria on selective media aquabides with an incubation period of up to 30 days can be seen based on the optical density of the growth of the two types of bacteria measured at λmaks 600 nm [3,9], according to Figure 3.

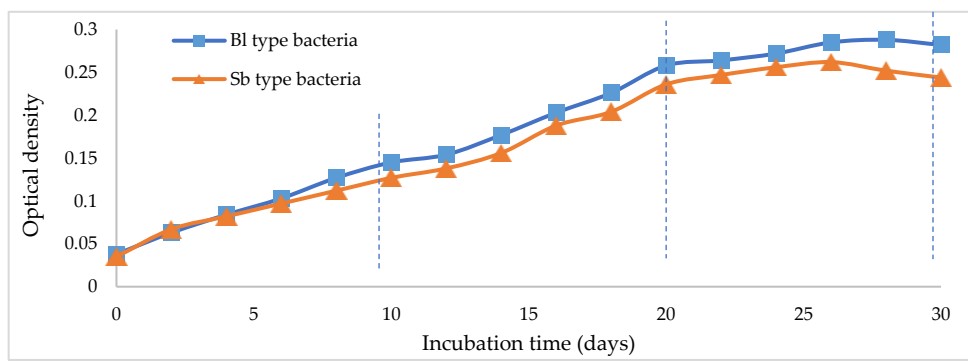

**Figure 3.** OD 600 values of both bacterial cells (Bl and Sb) in aquabides media without adding pyrene contaminants.

Based on the optical density (Figure 3), it was shown that Bl bacteria had a higher optical density than Sb bacteria. This situation indicates that the population and cell size of Bl bacteria is large, and growth is more aggressive than that of Sb bacteria. Changes in OD600, based on the incubation period, indicated the activity of both bacterial cell types (Bl and Sb). Figure 3 shows that the growth rate of Bl bacteria is relatively higher than that of Sb [4,6,19]. Thus, it can be predicted that the level and strength of degradation of Bl bacteria is dominant compared to Sb bacteria. In the incubation period, for the first 10 days, both bacteria appeared in the adaptation phase. The next 10 days, or an incubation period of 20 days, showed the bacteria in the cell division or multiplication phase. The incubation period of 30 days showed that bacterial cells have decreased activity [4,14].

A comparison of the optical density (OD) values of the two types of bacteria during the interaction with pyrene can be assumed to embody the biodegradation power of Bl and Sb bacteria against pyrene (Figure 4). It appears that the OD of Bl bacteria is higher than that of Sb bacteria. The difference in OD values begins to be seen at the 6-day interaction period, where the OD shown in the interaction of bacteria (Bl + pyrene) is higher than the OD value of the interaction of bacteria (Sb + pyrene). This situation continues until the interaction reaches 30 days; even the difference in OD values tends to get wider with the increase in interaction time. This indicator shows that the biodegradation activity of Bl bacteria against pyrene is more potent than that of Sb bacteria. In general, it can be said that both types of test bacteria have degradation activity against pyrene [30,73].

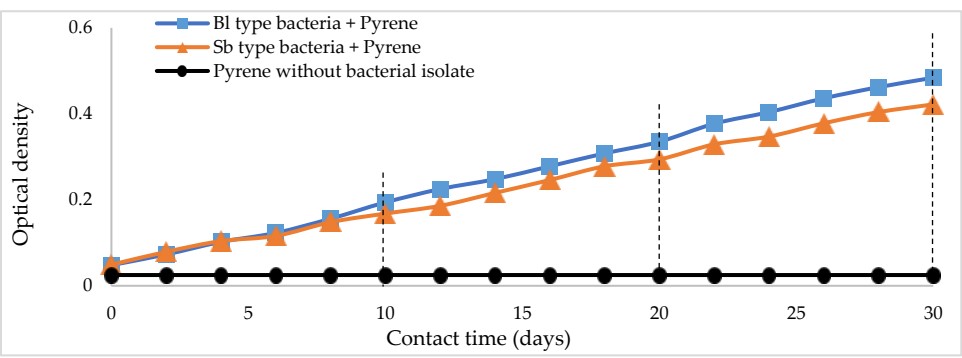

**Figure 4.** OD value of aquabides in each medium containing bacterial cells Bl + pyrene and Sb + pyrene.

Another indicator that shows that pyrene is degraded by Bl and Sb test bacteria according to several degradation parameters, including (1) increasing the temperature at the interval of 28–30 °C during the interaction period between days 10 to 24 and 26 to 30; (2) there was a change in the pH of the interaction medium at pH 6.64, which gradually decreased to pH 5.34 (the interaction medium was slightly more acidic). This condition occurred during the 8- to 30-day interaction period [24,33]; (3) gas bubbles in the interaction medium were seen on the eighth day of contact, and the population tended to increase with interaction time; and (4) the smell of fermentation from the interaction medium, observed on the 10th day of interaction until the 30th day of the measurement period. However, these four points cannot be used to distinguish the strength of the biodegradation activity between Bl and Sb bacteria in pyrene biodegradation [2,27,35].

The growth of Bl and Sb bacteria, respectively, suspended in aquabides media without the addition of pyrene contaminants (Figure 3) or with the addition of pyrene (Figure 4), there was no significant difference in growth. However, it was confirmed that these two types of bacteria (Bl and Sb) underwent growth in aquabides media exposed to pyrene contaminants, indicated by an increase in OD 600 values in each growth medium. The OD value of 600 in both distilled water media containing Bl + pyrene and Sb + pyrene bacteria, respectively (Figure 4), appeared to be higher than the OD 600 value for each aquadest medium which only contained Bl or Sb bacteria. This is due to the interaction between Bl/Sb bacterial cells and pyrene, resulting in increased turbidity in the media [12,19].

The increase in the value of OD 600 in distilled water media, suspended by Bl/Sb bacteria with the addition of pyrene contaminants, is thought to be caused by several factors, namely: (1) There was growth and division of the number of bacterial cells in each medium. (2) The biodegradation performance of Bl/Sb bacteria against pyrene produces biodegradation products in the form of simple organic compounds. (3) There is a parallel growth and division of bacterial cells with the formation of simple organic compounds resulting from biodegradation [2,22]. These three factors are measurement uncertainties, so they can be ignored because the measurement of OD 600 is only a qualitative analysis to ascertain whether Bl and Sb bacteria are tolerant to the presence of pyrene contaminants.

The biodegradation performance of Bl and Sb bacteria against pyrene was based on the results of GS/MS and FTIR analysis [4,90,91].

### 3.2. Comparison of the Biodegradation Performance of Test Bacteria

Analysis of the difference in the strength of the biodegradation of the two bacteria tested against pyrene is presented, based on aspects of the abundance of components and the number of peaks formed (Figures 5–7) and types and differences in organic compounds of degradation products (Tables 1 and 2). Similar studies using other marine sponge symbiont bacteria have been conducted [4,19,29,62]. That study observed the biodegradation performance of marine sponge symbiont bacteria against PAH components (naphthalene, anthracene, and pyrene) [4,6,12,66]. The red peak is the degraded pyrene component, while the other peaks are degradation products. This research initiated an analysis related to the comparison of biodegradation performance between bacteria isolated from seawater contaminated with the hydrocarbon component of the *Sphingobacterium* sp. strain 21 (Sb) versus the marine sponge symbiotic bacterium *Bacillus licheniformis* strain ATCC 9789 (Bl) against pyrene using a combination of GC/MS and FTIR analytical instruments [4,19,92].

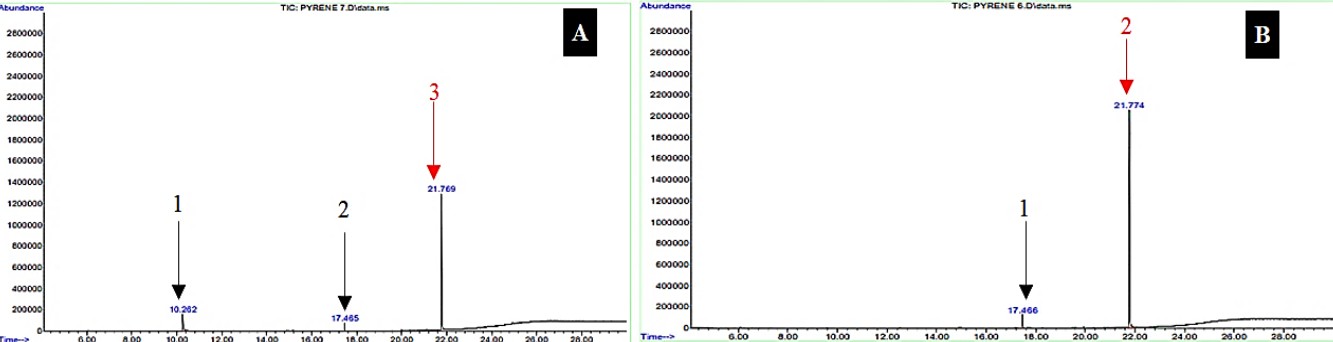

**Figure 5.** Comparison of peak abundances of biodegradation chromatograms, interaction time of 10 days; (**A**) chromatogram of degradation between bacterial isolate Bl and pyrene; (**B**) chromatogram of degradation between bacterial isolates of Sb and pyrene.

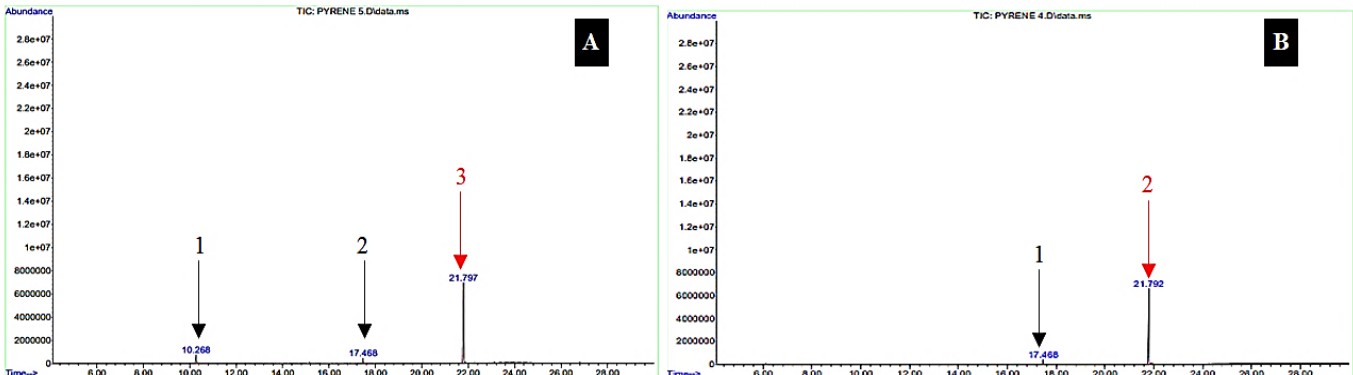

**Figure 6.** Comparison of peak abundances of biodegradation chromatograms, interaction time 20 days; (**A**) chromatogram of degradation between bacterial isolate Bl and pyrene; (**B**) chromatogram of degradation between bacterial isolate Sb and pyrene.

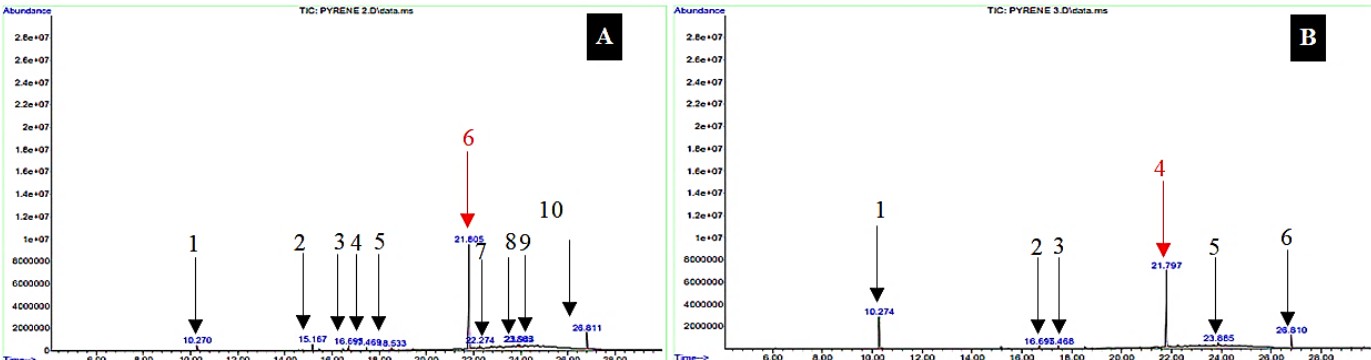

**Figure 7.** Comparison of peak abundances of biodegradation chromatograms, interaction time 30 days; (**A**) chromatogram of degradation between bacterial isolate Bl and pyrene; (**B**) chromatogram of degradation between bacterial isolate Sb and pyrene.

**Table 1.** GC/MS biodegradation reading data between *Bacillus licheniformis* strain ATCC 9789 (Bl) biodegradator bacteria against pyrene.

| Peak Number | Retention Time (Seconds) | Height Peak | Quality (%) | Compound Name |
|---|---|---|---|---|
| | | Contact time 10 days | | |
| 1 | 10.261 | 156,964 | 94 | Ethyl-methylazulene |
| 2 | 17.467 | 4366 | 45 | Meso-4,5-Dicyclohexyl-2 |
| 3 | 21.771 | 4,282,463 | 95 | Pyrene |
| | | Contact time 20 days | | |
| 1 | 10.267 | 730,588 | 94 | Dimethylazulene |
| 2 | 17.467 | 427,236 | 50 | 3-dimethyl-methanephosphonate |
| 3 | 21.796 | 3,809,710 | 96 | Pyrene |
| | | Contact time 30 days | | |
| 1 | 10.273 | 454,990 | 94 | Isopropyl azulene |
| 2 | 15.165 | 551,463 | 98 | Phenol, 2,6-bis(1,1-dimethylethyl) |
| 3 | 16.697 | 420,591 | 97 | Benzenemethanol |
| 4 | 17.467 | 282,380 | 50 | 5-methylbicyclo [3.2.0] heptan |
| 5 | 18.530 | 247,395 | 95 | Phenanthrene |
| 6 | 21.808 | 1,199,647 | 96 | pyrene |
| 7 | 22.271 | 277,763 | 48 | 1-Nonadecene |
| 8 | 23.885 | 208,195 | 52 | Eicosane |
| 9 | 23.947 | 222,869 | 56 | Tetrapentacontane |
| 10 | 26.812 | 1,507,974 | 87 | Terephthalic acid |

Note: Peak number according to GC/MS chromatogram (Figures 5–7) section A.

**Table 2.** Data of GC/MS biodegradation results between biodegradator bacteria *Sphingobacterium* sp. strain 21 (Sb) against pyrene.

| Peak Number | Retention Time (Seconds) | Height Peak | Quality (%) | Compound Name |
|---|---|---|---|---|
| | | Contact time 10 days | | |
| 1 | 17.467 | 127,964 | 52 | 1,3-dimethylbutyl phosphonate |
| 2 | 21.777 | 4,010,224 | 96 | Pyrene |
| | | Contact time 20 days | | |
| 1 | 17.467 | 407,716 | 52 | 2H-Tetrazole |
| 2 | 21.789 | 3,481,296 | 95 | Pyrene |
| | | Contact time 30 days | | |
| 1 | 10.273 | 2,826,324 | 94 | Dimethyilazulene |
| 2 | 16.698 | 261,361 | 97 | Benzenemethanol |
| 3 | 17.467 | 263,235 | 52 | 2H-Tetrazole |
| 4 | 21.796 | 2,788,232 | 96 | Pyrene |
| 5 | 23.883 | 229,305 | 58 | Tricosane |
| 6 | 26.812 | 1,144,755 | 87 | Terephthalic acid |

Note: Peak number according to GC/MS chromatogram (Figures 5–7) section B.

The difference percentage of pyrene that did not undergo biodegradation and the components organic compounds of biodegradation products are shown in Figures 8 and 9. It includes functional groups of organic compounds of biodegradation products (Figures 10 and 11). These three indicators provide qualitative and quantitative data on the strength of the biodegradation of the two types of bacteria against pyrene [40,48,61].

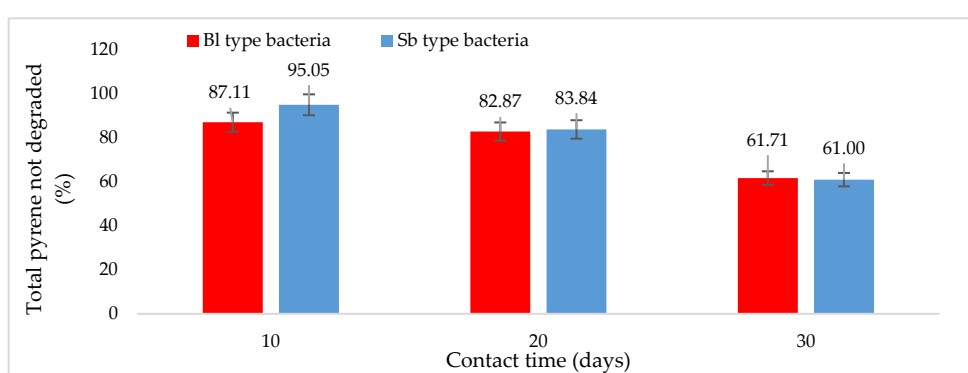

**Figure 8.** Comparison of the percentage of pyrene components as a substrate not degraded by Bl and Sb bacteria based on interaction time.

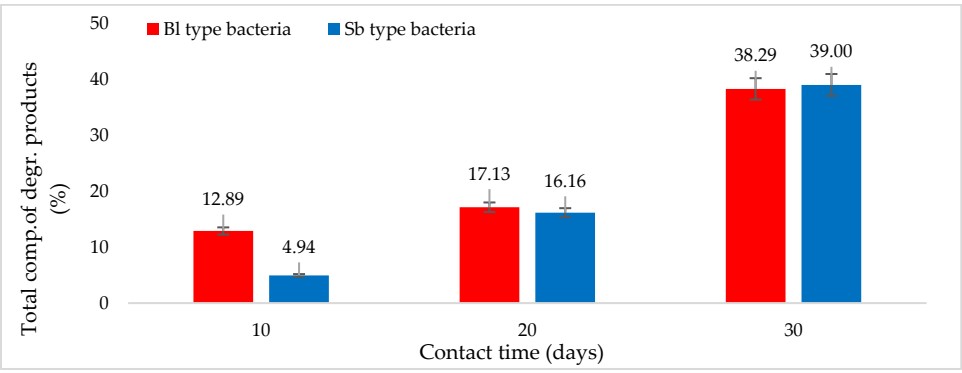

**Figure 9.** Comparison of the percentage of the total components of bacterial biodegradation products of Bl and Sb types to pyrene based on interaction time.

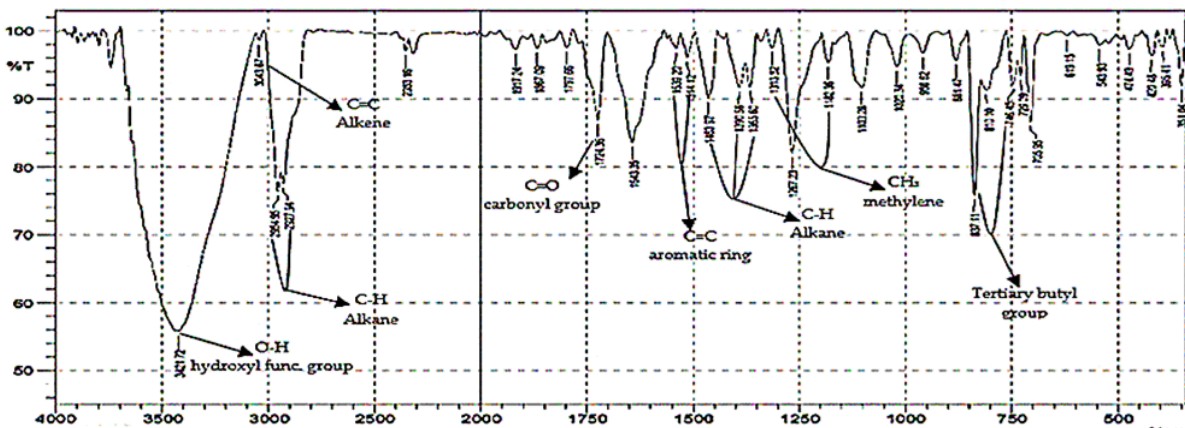

**Figure 10.** FTIR spectrum showing functional groups of organic components in the biodegradation performance of *Bacillus licheniformis* strain ATCC 9789 (Bl) against pyrene after 30 days of interaction.

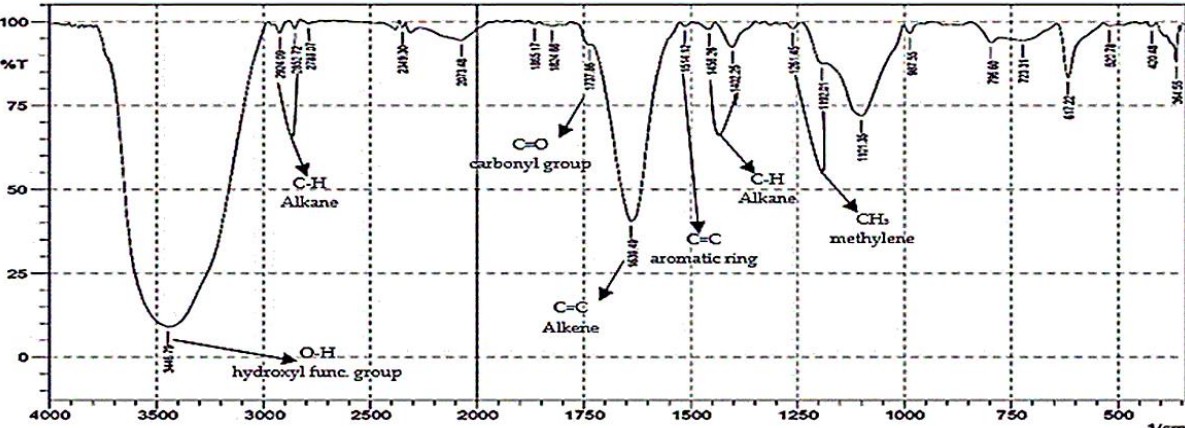

**Figure 11.** Spectrum showing functional groups of organic components in the biodegradation performance of *Sphingobacterium* sp. strain 21 (Sb) against pyrene after 30 days of interaction.

Based on a comparison of GC/MS chromatograms at 10-day contact of biodegradation of pyrene components, there were three peaks (Figure 5A) resulting from the interaction of Bl bacteria with pyrene, and there were two peaks (Figure 5B) in the interactions between Sb bacteria and pyrene, indicating that B1 bacteria more potently reduced aggressiveness of pyrene compared to Sb bacteria [94,95]. The GC/MS chromatogram at the 20-day interaction period between Bl and Sb bacteria against pyrene (Figure 6A,B) showed a relatively similar appearance to that of the 10-day interaction, especially the number of peaks [4,90,92].

However, the peak height indicated that the pyrene component decreased more sharply than in during the first 10 days of interaction. On the other hand, the components suspected of being biodegradation products experienced a slight increase in peak height, indicating an increase in the components of degradation products [38,94]. Chromatograms for the 30-day interaction between Bl and Sb bacteria with pyrene (Figure 7A,B) exhibited significant changes, especially in the number of new peaks formed, indicating that the peak height of the pyrene component experienced a sharp decrease. Ten peaks were identified in the chromatogram of the interaction of Bl bacteria with pyrene (Figure 7A), indicating that there were nine components of the bacterial biodegradation product type Bl.

In contrast, in the interaction of bacteria Sb with pyrene (Figure 7B), only six peaks were seen, indicating five identified peaks with the component of the product of bacterial biodegradation of Sb. Numbers and arrows in red (Figures 5–7) indicate the pyrene component as a substrate that undergoes degradation [7,40,74]. These results showed

that Bl-type bacteria isolated from the marine sponge *Auletta* sp. did not have a more aggressive biodegradation activity against pyrene than Sb-type bacteria isolated from seawater. [36,41,96]. The biodegradation dynamics of the two types of bacteria tested against pyrene are shown in Tables 1 and 2.

Potential formation of pyrene derivative products as a result of bacterial biodegradation in biodegradation was studied for bacteria types Bl and Sb. First, the pyrene molecule was formed by combining four benzene molecules. The benzene ring breaks when pyrene undergoes biodegradation through an oxidation reaction mechanism in one of the benzene structures in several stages through oxidation metabolism [3,94]. The exact process can occur in the second benzene molecule until the reaction ends with one benzene molecule. Two benzene molecules are broken apart at this stage, leaving two benzene molecules intact, possibly forming a naphthalene molecule. This assumption can be proven validly, but it takes a long series of extensive research, a super and accurate analysis with the support of a complete analysis using NMR [94,95].

Second, the use of glassware and analytical instruments during the sample preparation process leads to the possibility of equipment contamination with naphthalene components due to human error. If the second assumption is valid, this is an oversight from our analysis work, but it was emphasized that work was carried out correctly and according to the procedure during the testing process. This can also be seen in the data in Table 2, as no naphthalene was detected even though the procedures we carried out were identical and simultaneous [4,96].

The results of the biodegradation of B1 bacteria against pyrene were indicated by the GC/MS readings (Table 1). Several benchmarks indicate that there was biodegradation of pyrene as a substrate by type Bl biodegradator bacteria, including: first, the decrease in the peak height of the pyrene component, which is analogous to the decrease in pyrene concentration due to changes in structure, decomposition, degradation, and reduction in components as a result of bacterial activity type Bl [1,4,6,8,91]. Second, the decrease in the percentage area of the pyrene component is a sign that the concentration of pyrene reduces as the interaction time between Bl bacteria and pyrene increases. Third, the percentage of the total composition of the biodegradation product increased with contact time between Bl bacteria and pyrene. Furthermore, the number of components formed tends to increase following the increase in interaction time [2,96].

However, the number of these components is not entirely seen as the end product of degradation. The final product of bacterial pyrene biodegradation cannot be ascertained with a constant. Components of biodegradation products (Tables 1 and 2) can be divided into two categories: First, components with product quality ≥ 90% (similarity level indicated by reference or GC/MS library) are assumed to be the final products of biodegradation according to contact time. Second, components with content of <90% are intermediate products [90,95]. The final product of biodegradation is still very likely to change towards a simple organic compound in the form of methyl if the interaction time is added as long as it is believed that bacteria are still working (the biodegradation process will continue). The components of the biodegradation products (Tables 1 and 2) can be said to be intermediate products [94,95,97].

Certain bacteria can carry out the biodegradation of compounds containing carbon, where bacteria can convert carbon into energy, so it is assumed that there are almost no constant and permanent biodegradation products. These components will continue to change until they reach simple organic compounds because bacteria carry out their biodegradation function act as enzymes, so that the biodegradation of pyrene as a substrate may be a fermentation reaction, and then it cannot be categorized as the final product of bacterial biodegradation because the component may be a transition product [19,62,66,98].

The analysis of the biodegradation process for Bl bacteria against pyrene, according to the GC/MS reading data (Table 1), was identical to the biodegradation data for Sb bacteria against pyrene (Table 2). Comparative analysis of the biodegradation strength between *Bacillus licheniformis* strain ATCC 9789 (Bl) and pyrene based on interaction time

showed the symbiont bacteria of the marine sponge *Auletta* sp [4,12,19]. The biodegradation power is relatively balanced compared to *Sphingobacterium* sp. strain 21 (Sb) isolated from marine water contaminated with PAHs. However, it does not necessarily mean that marine sponge symbiont bacteria have weaker biodegradability against PAH components than bacteria isolated from marine water contaminated with hydrocarbon component pollutants [15,18,98]. General conclusions regarding the biodegradation strength of marine sponge symbiont bacteria compared to bacterial isolates from seawater contaminated with PAHs require a comprehensive bacterial investigation and analysis [48,71,97].

*3.3. Biodegradation Performance*

Comparative analysis of the biodegradation strength between Bl and Sb bacteria against pyrenes was based on interaction time (Figure 8). In general, it can be seen in the total pyrene that it was not degraded by the test bacteria.

The results of the analysis of the biodegradation activity of the tested bacteria on the pyrene component showed no significant difference in the strength of the biodegradation. However, there were differences in the total degraded pyrene component at several times of observation and measurement. The dominance of the biodegradation power of B1 bacteria against pyrene was seen in the contact phase for the first 10 days. In the second 10-day contact phase, it appears that the biodegradation strength of the two types of bacteria tested is relatively balanced [2,96].

Even in the third 10-day contact phase, the biodegradation strength of Sb bacteria appears to outperform Bl type bacteria (Figure 8). These results have implications for the total biodegradation products predicted to follow a relatively undifferentiated pattern.

Aspects of the percentage analysis of the total components of the biodegradation product as a result of the work of the two types of bacteria tested against pyrene (Figure 9) showed a similarity in the path of biodegradation strength [7,14]. The percentage of total biodegradation products of Bl test bacteria in the first 10 days of contact phase was greater than that of Sb's total biodegradation products. However, in the second 10 days of contact, the percentage of total biodegradation products between the two test bacteria showed similar results. Even at the third phase of contact, the 30th day of the interaction period, the percentage of total biodegradation products of Sb type bacteria was more significant than that of Bl type bacteria [5,11,94,95].

An overview of the energy aspects of most molecular vibrations relates to the infrared region. Molecular vibrations can be detected and measured in the infrared spectrum. The results of the FTIR spectra analysis showed that after the interaction between the pyrene component and the suspension of the test bacteria, the pyrene component decomposed into simple organic compounds, which could be analyzed based on the wavenumbers of the functional groups according to the FTIR chromatogram shown by the Bl and Sb test bacteria [6,12,68,99].

The results of FTIR analysis (Figure 10) showed one of the degradation products of simple organic compounds in the form of phenol, alcohol monomers, hydrogen-bonded alcohol, a carboxylic acid group, carboxylic acid hydrogen bonds, or aromatic carbon-hydrogen bonds. Absorption in the range 3200–3600 $cm^{-1}$, specifically 3444.87 $cm^{-1}$, indicates the presence of the -OH functional group. The absorption at 2958.80 and 2929.87 $cm^{-1}$ showed a peak with a characteristic shape of the absorption region of C–H alkanes. The absorption area of 1610–1680 $cm^{-1}$, precisely at the peak of 1639.49 $cm^{-1}$, shows a typical shape representing C=C alkenes [95,99].

The absorption area is 1500–1600 $cm^{-1}$ and shows the typical shape of the aromatic C=C bond. The peak of the absorption indicates the presence of aromatic cyclic bonds. The absorption area is 1050–1300 $cm^{-1}$, indicating the presence of the C–O functional group, precisely at the peaks of 1093.64 $cm^{-1}$; 1192.01 $cm^{-1}$; and 1261.45 $cm^{-1}$, suggesting the absorption of compounds that have a –OH (hydroxyl) functional group, each indicating a compound of alcohol, ether, carboxylic acid, and ester groups. Aromatic rings with C–H bonds appear at absorption in the range of 690 to 900 $cm^{-1}$, while at peaks of 40.67 and

794.67 cm$^{-1}$, they indicate the presence of aromatic cyclic bonds [4,21,98,99]. These results indicate the suitability of the components of simple organic compound biodegradation products of type B1 bacteria against pyrene (Table 1).

FTIR spectra analysis (Figure 11) shows the components of simple organic compounds, the result of pyrene biodegradation of Sb bacteria. In general, it can be said that these simple organic components are the result of the decomposition of pyrene components in the form of alcohol group organic compounds. In phenol, groups of alcohol monomer, hydrogen-bonded alcohol, phenol carboxylic acid monomer, and carboxylic acid hydrogen bonds will be absorbed in about 3200–3600 cm$^{-1}$ and 3446.79 cm$^{-1}$, indicating the presence of the –OH functional group. Absorption at 2924.09 and 2852.72 cm$^{-1}$ showed a typical peak shape identical to the C–H absorption region of alkanes [38,99,100].

Based on the data above, it is generally assumed that the biodegradation mechanism of PAHs by both types of bacteria (Bl and Sb bacteria) is estimated to be relatively the same as the biodegradation process of other types of PAHs, such as anthracene and phenanthrene. The rate of PAH metabolism by microorganisms depends on the number of aromatic rings. This biodegradation mechanism also has a metabolic pathway similar to pyrene using *Mycobacterium* sp. PYR-1[45,56,88], through the oxidation pathway, breaks one benzene molecule to form a new molecule in the form of a carboxylic compound and frees H$_2$O molecules. It can be seen that there are limiting factors that inhibit the biodegradation process of microorganisms against PAH components. So that biodegradation is not complete or leaves an aromatic benzene molecule. Thus, the biodegradation process converts carbon elements into energy through metabolic pathways, and the oxidation reaction proceeds slowly or even stops altogether [19,23,66].

Based on the above variables and assumptions regarding the mechanism of pyrene biodegradation, it can be stated that the bacterial isolate of *Bacillus licheniformis* strain ATCC 9789 (Bl) has a relatively balanced biodegradation power against pyrene when compared to *Sphingobacterium* sp. strain 21 (Sb) [4,6,97,101]. The biodegradation performance of the two types of bacteria (Bl and Sb) tends to increase along with the increase in contact time with pyrene contaminants [12,19]. The mechanism of pyrene biodegradation using two different types of bacteria (Bl and Sb) followed a relatively similar pathway, although these bacteria were isolated from different sources [1,102]. The product of biodegradation of hydrocarbon components by marine bacteria is in the form of simple organic compounds.

## 4. Discussion

The study of qualitative and quantitative aspects of the biodegradation performance of the two isolates (Bl and Sb) showed contradictory results. The qualitative analysis of the biodegradation performance of Bl bacteria against pyrene showed a stronger aggressiveness than Sb bacteria. This is based on the GC/MS chromatogram and the number of biodegradable components after 30 interactions which reached 10 components (Figure 7A and Table 1) [17,94]. These results are confirmed by the FTIR spectrum, which appears to be more complex, including visible functional groups, as a manifestation of the biodegradation products in the form of simple organic compounds (Figure 10). Qualitative analysis of the biodegradation performance of Sb, according to the GC/MS chromatogram, identified only six components (Figure 7B and Table 2) [18,96].

Similarly, the FTIR spectrum appears simpler (Figure 11). The results of the quantitative analysis found that for the biodegradation performance of the two types of bacteria against pyrene, it appears that Sb bacteria are relatively stronger than Bl bacteria [21,99]. The percentage of pyrene components that were not degraded by Bl bacteria (61.71%) was relatively higher than that of Sb bacteria (60.00%) (Figure 8). This suggests that the biodegradation performance of Sb bacteria is relatively higher than that of Bl bacteria. These data are corroborated by the percentage of bacterial biodegradation performance of Sb (39.00%), slightly higher than the performance of bacteria Bl (38.29%) (Figure 9) for pyrene components [3–5,11,23]. Statistical analysis showed no difference in the biodegradation performance of the two types of bacteria (Bl and Sb). This result is influenced by the

adaptability and biodegradation mechanism of two types of bacteria that can exert on the pyrene component.

Based on the data of biodegradation parameters such as fermentation reactions, in the form of OD values, changes in pH, interaction temperature, and the presence of gas bubbles and fermentation odors combined with GC/MS and FTIR data, it can be said that both types of bacteria can carry out the biodegradation function of pyrene [4,89]. The biodegradation products are organic compounds, for example, alcohol, aldehyde, and carboxylic acids. The level of performance of the biodegradation of the two types of bacteria (Bl and Sb) against pyrene is relatively tiny [36,95,99]. The biodegradation mechanism is an oxidation reaction, namely, the entry of -OH molecules, followed by the breakup of one benzene molecule, which marks the change in the structure of pyrene into a carboxylic acid product. Ideally, biodegradation can run continuously until the final product is reached, namely, benzanoate molecules in the form of cinnamate and pinacol products, provided that the interaction time is extended and it is believed that there are still bacteria in the working biodegradation reactor. This process is called the final reaction or the termination step of biodegradation [27,31,39,88].

The illustration in Figure 4 shows a symbiont of the marine sponge *Auletta* sp., a group of Bacillus bacteria, *Bacillus licheniformis* strain ATCC 9789 (Bl), with relatively similar pathways for metabolism or pyrene biodegradation using *Sphingobacterium* sp. strain 21 (Sb) bacteria isolated from marine water contaminated with PAHs. In general, it can be said that the biodegradation mechanism of PAHs is similar to the biodegradation process of other types of PAHs, such as anthracene and phenanthrene [39,46,102].

The rate of metabolism of PAHs by microorganisms depends on the number of aromatic rings. This biodegradation mechanism also has similar metabolic pathways to pyrene using *Mycobacterium* sp. PYR-1 [45,56,88] through the oxidation pathway of breaking one benzene molecule so that a new molecule is formed in the form of a carboxylic compound and frees $H_2O$ molecules. According to the results of the analysis (Figures 5–7, 10 and 11, as well in Tables 1 and 2), there is a limiting factor that inhibits the biodegradation process of microorganisms against PAHs components so that the biodegradation is not complete or leaves the aromatic benzene molecule [88]. Thus, the biodegradation process converts carbon elements into energy through metabolic pathways, and the oxidation reaction proceeds slowly and even stops completely [19,23,66].

The results of this study are expected to be developed for screening of other types of bacteria from various sources that can biodegrade hydrocarbon components, especially PAHs [94,95]. The achievements of this research also open up opportunities for using these bacteria in the bioremediation of other types of pollutants, such as pesticide residues, heavy metals, and microplastics, so that the goal of formulating crystalline carbonoclastic bacteria can be realized [12,21,64]. The aim is a form of environmental protection against the threat of contamination by toxic components.

## 5. Conclusions

Some of the findings obtained from this study are summarized in several conclusions: The two types of test bacteria (Bl and Sb) can degrade the pyrene component. A qualitative study based on the number of components and FTIR spectrum showed that *Bacillus licheniformis* strain ATCC 9789 (Bl) had the same strong biodegradation performance as *Sphingobacterium* sp. strain 21 in pyrene biodegradation. Quantitative analysis showed that the pyrene component's biodegradation performance of Sb bacteria was relatively stronger than that of Bl bacteria. The total bacterial biodegradation product type Sb (39.00%) was slightly higher than the type Bl biodegradation product (38.29%) achieved during the 30-day interaction period. There is no statistically significant difference in the biodegradation performance of these two types of bacteria. It means that it can be ignored. The biodegradation products of the two test bacteria (Bl and Sb) against pyrene were simple organic compounds with alcohol and carboxylic acid groups. The biodegradation perfor-

mance of the two types of bacteria tested against pyrene followed the same path, namely, carbon metabolism as an energy source through oxidation reactions.

**Author Contributions:** Conceptualization, I.M., R.A., A.A. and K.N.; methodology, I.M., M.P., K.N., and R.A.; software, A.A., R.A., A.R. and E.S. (Endang Susianingsih); validation, I.M., N.N. (Nurhidayah Nurhidayah), R.A., A.A. and I.A.K.K.; formal analysis, I.M., K.N., R.A. and S.S. (Sahabuddin Sahabuddin); investigation, I.M., N.N. (Nurbaya Nurbaya), R.A. and H.H.; resources, E.S. (Early Septiningsih), E.A.H., A.A. and K.N.; data curation, I.M., K.K., K.N. and M.P.; writing—original draft preparation, I.M., K.N., A.A. and R.A.; writing—review and editing, A.A., K.N., N.N. (Nurhidayah Nurhidayah) and E.S. (Endang Susianingsih); visualization, S.S. (Sahabuddin Sahabuddin), K.N., E.A.H. and E.S. (Early Septiningsih); supervision, I.M., R.A., S.S. (Suwardi Suwardi), A.A. and I.A.K.K.; project administration, I.M., H.H., A.R. and A.A.; funding acquisition, I.M., N.N. (Nurbaya Nurbaya), S.S. (Suwardi Suwardi) and K.K. All authors have read and agreed to the published version of the manuscript.

**Funding:** This research received external funding by the Ministry of Research, Technology and Higher Education/National Research and Technology Agency, for granting grant funds in 2021 in the applied research scheme, according to the Master Contract number: 315/E4.1/AK.04.PT/2021, and Number: 1868/E4/AK.04/2021, dated 7 June 2021.

**Data Availability Statement:** No data availability statement.

**Acknowledgments:** Thanks are conveyed to the Ministry of Research, Technology and Higher Education/National Research and Technology Agency for granting grant funds in 2021 in the applied research scheme and to the management of the Biochemistry Laboratory, Department of Chemistry, Faculty of Mathematics and Natural Sciences, Hasanuddin University, and the Microbiology Laboratory of BRPBPPP Maros, South Sulawesi, and the Forensic Laboratory of the South Sulawesi Police.

**Conflicts of Interest:** The authors declare no conflict of interest.

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
