# Peer review of "Comparison of Pyrene Biodegradation Using Two Types of Marine Bacterial Isolates"

_sustainability, doi:10.3390/su14169890_

Round 1
Reviewer 1 Report
This manuscript is clearly written and relevant. The analytical techniques are appropriate and well described. Results are also appropriate. Three hyphens in the first paragraph (Lines 39, 41, 42) should be removed.
Author Response
Dear
Reviewer 1
Thanks for the peer reviewers that have been done on our manuscript. Excellent response, because the reviewer has pointed out some mistakes in writing words. The revised manuscript and the author's response are attached.
Best regards,
Ismail Marzuki and Friends

Reviewer 2 Report
· The english of the manuscript should be reworked and revised.
· Some words should be revised like:
· Line 131 : what the meaning of PA, PA ?
· Line 134-138: the description of the GC program should be reformulated and revised (temperature 350°C of what?)
· Line 151 : distance between the two sampling points is ± 3.45 Nm or Km ?
· It is recommended to put the error bars in the the graphic of the figure 3!
· Lines 213-15 : phrase (changes in optical density……) should be reformulated….
· All graphics and graphics should be re-worked in a clear form.
· What about the toxic level of chemical compounds resulted from the biodegradations of PAHs ? I am not sure that these molecules are also safe for the environment. It is important to comment this point!
· The percentage of biodegradation (38-39%) is considered good for the safety of the environment? and what about the 60% of PAHS remained in the environment!
· It is important to indicate the method used in the calculation of the percentage of biodegradation ?
Author Response
Dear
Reviewer 2
Thanks to the peer reviewers who have done on our manuscript. Excellent response, because the reviewer has pointed out some mistakes in writing words, inaccurate phrases, and graphic/image improvements.
The revised manuscript, the author's response, and the author's feedback on several reviewer statements require a more detailed explanation. Revised manuscript and author's response, as attached.
Greetings,
Ismail Marzuki and Friends

This manuscript is a resubmission of an earlier submission. The following is a list of the peer review reports and author responses from that submission.
Round 1
Reviewer 1 Report
The paper from Markuzi and collaborators describes the evaluation of two strains for pyrene biodegradation. To this reviewer opinion, this manuscript is not suitable for publication in Sustainability journal.
The manuscript is poorly written, the English style needs a revision, and it contains continue repetition of the same sentences. The authors state the isolation site of the bacteria several times and yet, they never clarify if they performed the isolation or not. Also, how come the bacteria isolated from a marine sponge corresponds to a Bacillus strain present in the ATCC bank and isolated from milk?
The M&M lacks tremendous amount of details, the protocols are not clear, and even the media composition is not stated.
The results are quite preliminary and do not go into details as the authors state. They merely report a growth curve and one graph, without any abiotic control reporting the pyrene degradation of the strains. All the other GC-MS analysis (for which there is no protocol) are of low relevance.
The authors also state that this is the first paper to utilize bacteria isolated from sponges (not clear if they did) for PAHs degradation. This is simply untrue, and below there is just the most recent example I could find:
https://www.ncbi.nlm.nih.gov/pmc/articles/PMC8624637/
Reviewer 2 Report
- Title.
“Performance” is not usual word for description of bacterial activity. I recommend an emphasis on degradation of pyrene: “Pyrene Biodegradation by Two Marine Bacteria”.
- Abstract.
Abstract is an essential part of article because it (i) saves time of potential readers, (ii) pays special attention to the main results. Thus, please,
- remove lines 19-21 to the “Discusson” section;
- present the whole abstract in more simple and clear style. For example, lines 30-33: “It was concluded that there was no significant difference in biodegradation performance between Bl and Sb bacteria on for pyrene. Both types of bacterial isolates from different sources can carry out the function of biodegradation of pyrene” must be reduced to: ”There was no significant difference in the pyrene biodegradation between Bl and Sb bacteria.”
- Introduction.
The introduction contains redundant descriptions of: 1) characteristics of the group of polyaromatic hydrocarbons in general; 2) characteristics of aquatic bacteria in general; etc. These descriptions are in the nature of educational popular literature. Please rewrite the full “Introduction” section. The content must be limited with the following statements:
- pyrene is toxic aromatic compound,
- pyrene is a known pollutant for various ecosystems, including marine environment,
- pyrene can be degraded by various microorganisms (by the way, please, never name bacteria “materials”: this word might be applied to any organism just if we discuss them as a non-living mass),
- the presented article is dedicated to study if two bacterial strains of marine origin could degrade pyrene.
Thant’s enough – any introduction must be short and clear. Never overload it with references (87 references!): it is just an introduction but not a review.
- Materials and Methods.
Main aim of the section “Materials and Methods” is to provide any reader with possibility to repeat / to check the presented study. Please reduce the section according to this point of view.
4.1. “Materials” – rename this subsection to “Bacterial strains” or “Used bacterial strains”.
Reduce the whole subsection to: “Two bacterial strains, namely Bacillus licheniformis ATCC 9789 (Bl) isolated from the marine sponge Auletta sp. and Sphingobacterium sp. 21 (Sb) isolated from marine water were used in experiments. These bacteria were selected by their biodegradation potential of PAHs in previous studies [3,5,92].” (Lines 141-143, lines 206-207.)
4.2. Subsection “Sampling” is outside the scope of experiments and does not contain any information the reader needs. Please eliminate this subsection.
4.3. Subsections “Sample Preparation” and “Performance of bacteria and biodegradation products” must be united into one subsection “Description of experiments”. This subsection must be shortened. For example, lines 171-175 must be eliminated.
4.4. Please add one subsection more: “Statistics”. The statistical analysis is very essential. For example, conclusions contain the sentence “The total percentage of bacterial biodegradation products of Sb type (39.00%) was slightly higher than that of Bl type biodegradation products (38.29%) achieved during the interaction period of 30 days”. “Slightly higher” - is it statistically significant or not? As well, this analysis is a measure for the “human errors” too: “Second, the use of glassware and analytical instruments during the sample preparation process leads to the possibility of equipment contamination with naphthalene components due to human error” (lines 341-343).
- Results.
5.1. Subsection “Morphological analysis “. The morphology of bacteria can be left in the article only if it changes depending on the biodegradation of pyrene. Otherwise, it is outside the scope of this article and should be excluded.
5.2. Style and excess content. Extensive editing of English language and style required. An example is (lines 331-332): “Potential formation of pyrene derivative products as a result of bacterial biodegradation in biodegradation was using was studied for bacteria types Bl and Sb.” Please rewrite the new results and present them in a more clear description.
- In nutshell, the article must be rewritten, shortened, and presented with a distinct description: what results are new and why they could be of interest.

Reviewer 3 Report
This study investigated the feasibility of PAH biodegradation by two types of marine bacteria. The topic is of potential interest for readers. I have several comments for authors' consideration:
1- The English of this paper should be improved. For example:
L22: it should be “the study aims” instead “the study aimed”
L22-23: it should be “to compare the capacity of PAH biodegradation by two types of bacteria isolated from different sources” instead “to compare the biodegradation power of two types of bacteria isolated from different sources against PAHs”
L26: it should be “The capacity of pyrene degradation by BI bacteria” instead “the aggressiveness of biodegradation of Bl bacteria against pyrene”
Authors should use the present tense to announce results
2- There should be full names for the abbreviations in their first appearances, for example: PAH, BI, Sb, GC/MS, FTIR…
3- Please add a figure of the Molecular structure of pyrene and some typical polycyclic aromatic hydrocarbons (PAHs)
4- The introduction part can be improved by:
adding more details about the mechanisms involved in the PAH biodegradation
adding a figure of the Molecular structure of pyrene and some PAHs
removing repeated ideas
5- The "Materials and Methods" part is unclear. It is difficult to follow, and it is difficult to understand the experimental approach carried out. Authors should rewrite this part as follows:
- Collection of samples or sampling sites
- Isolation of bacteria
- Screening of the isolated strains
- Culture conditions and growth monitoring of the selected stains
- Morphology analysis of the selected strains
- Biodegradation experiment
- Extraction of pyrene for GC/MS and FTIR analysis
6- The authors need to present a microscopic image or a photo of an isolated strain in order to support the bacterial morphology description
7- The quality of Figures 3,4, 8 and 9 need to be improved.
8- L225: What is the selective medium used?
9- The description of the results does not help in the understanding of this manuscript. Authors must be precise and well organized according to each idea or result. For example, in the page 6, the authors present the growth results of the two isolated bacteria. However, in the description of these results, we find that the authors discuss the capacity of PAH degradation by these two bacteria.
10- Better growth does not imply better biodegradation efficiency. If the OD increases, the rate of biodegradation may increase. However, to measure the effectiveness, it is necessary to divide the biodegradation rate by the OD.
11- Given the GC/MS and FTIR data, what are the authors' predictions about the Biodegradation mechanism and the enzymes involved in this process?
12- Figures 8 and 9: Authors should present these results as total XX/OD versus time
13- Abstract, discussion and conclusions should be modified after this major revision